# Light-driven nucleation, growth, and patterning of biorelevant crystals using resonant near-infrared laser heating

**Marloes H. Bistervels[1], Balázs Antalicz [1], Marko Kamp[1], Hinco Schoenmaker[1] & Willem L. Noorduin [1,2]**

Spatiotemporal control over crystal nucleation and growth is of fundamental interest for understanding how organisms assemble high-performance biominerals, and holds relevance for manufacturing of functional materials. Many methods have been developed towards static or global control, however gaining simultaneously dynamic and local control over crystallization remains challenging. Here, we show spatiotemporal control over crystallization of retrograde (inverse) soluble compounds induced by locally heating water using near-infrared (NIR) laser light. We modulate the NIR light intensity to start, steer, and stop crystallization of calcium carbonate and laser-write with micrometer precision. Tailoring the crystallization conditions overcomes the inherently stochastic crystallization behavior and enables positioning single crystals of vaterite, calcite, and aragonite. We demonstrate straightforward extension of these principles toward other biorelevant compounds by patterning barium-, strontium-, and calcium carbonate, as well as strontium sulfate and calcium phosphate. Since many important compounds exhibit retrograde solubility behavior, NIR-induced heating may enable light-controlled crystallization with precise spatiotemporal control.

Controlling crystallization at the right place and at the right time is at the core of how organisms organize minerals into complex architectures with superb performance[1–6]. For instance, biominerals of calcium carbonate and calcium phosphate crystallize into hierarchical structures such as dental enamel[7], and nacre for creating materials with high resilience and hardness[8], and crystallographically well-ordered calcite crystals in starfish tentacles for compounded vision[9–11]. Gaining such spatiotemporal control over crystallization offers opportunities for probing, steering, and understanding crystallization phenomena in organisms, and enables the development of artificial materials with advanced functionalities[12–18]. Already, many techniques such as self-assembled monolayers[19,20], microfluidic devices[21,22], confined spaces[23,24], (anisotropic) templates[25–27], topographical substrates[28,29], or additives[30,31], have been developed to achieve control over crystallization in either space or time, but controlling both simultaneously remains challenging[32].

From this perspective, light-induced reactions offer the potential to gain spatiotemporal control over crystallization by modulating light intensities with (sub)micrometer resolution and microsecond dynamics[33,34]. Photochemical reactions have been used to control assembly and nucleation of colloidal and mineral systems by using for instance photoswitches[35,36], photoacids/bases[37,38], and photodecarboxylation[39], that steer local interactions or local gradients. However, these photochemical reactions require complicated chemical mixtures of photoreactants, photoproducts, and stabilizing agents such as surfactants; all of which can inflict undesired disturbance of both crystal nucleation and growth[40]. Alternatively, non-photochemical processes have been explored to

[1]AMOLF, 1098 XG Amsterdam, The Netherlands. [2]Van't Hoff Institute for Molecular Sciences, University of Amsterdam, Amsterdam 1090 GD, The Netherlands. ✉ e-mail: noorduin@amolf.nl

induce crystallization in solutions, thin films, and glasses[41–43]. But the need for complicated pulsed lasers set-ups, special crystallization media, or large minimal irradiation area (typically mm²) in combination with the inherent stochastic nature of nucleation has so far restricted the application potential and the spatial control[44]. Moreover, the limited understanding of the underlying principle has so far prevented further rational extension of these methods[45]. Unlocking the full potential of light-controlled crystallization processes thus requires an approach to overcome the need for complicated mixtures that disturb crystal nucleation and growth, while at the same time offering generalizable principles with precise spatiotemporal control.

From this perspective, near-infrared (NIR) laser light provides opportunities. NIR laser light enables local heating of water by resonating with the first overtone of the O-H stretch vibration (Fig. 1), and is routinely utilised to manipulate biological processes such as protein kinetics and DNA folding[46,47], and NIR light-triggered release of payload from polymeric capsules[48]. In contrast, we here explore NIR laser light to induce crystallization with precise spatiotemporal control while

circumventing the need to use special crystallization media or additives. Our strategy for controlling crystallization using NIR is motivated by two key insights: (i) many biorelevant minerals (e.g., calcium carbonates[49] and calcium phosphates[50]), and many technologically important crystals (e.g., lead halide perovskite semiconductors[51]) exhibit a retrograde solubility, i.e. the solubility of these compounds decreases for increasing temperatures. (ii) NIR laser light has been shown to locally heat up water with 50 °C in a 10 micrometer diameter light spot within milliseconds, suggesting great potential for locally increasing the supersaturation and thereby inducing crystallization of retrograde soluble minerals with precise spatiotemporal control without any need of additives[47].

Motivated by this analysis, we develop an NIR optical setup to locally induce crystallization of compounds with an inverse solubility. By tuning the crystallization conditions and light settings we demonstrate the following levels of spatiotemporal control in user-defined micrometer patterns for metal carbonates: (1) writing of CaCO₃ crystals in microscopic user-defined patterns; (2) refining the resolution for nucleation and growth of single crystals; (3) extending NIR induced

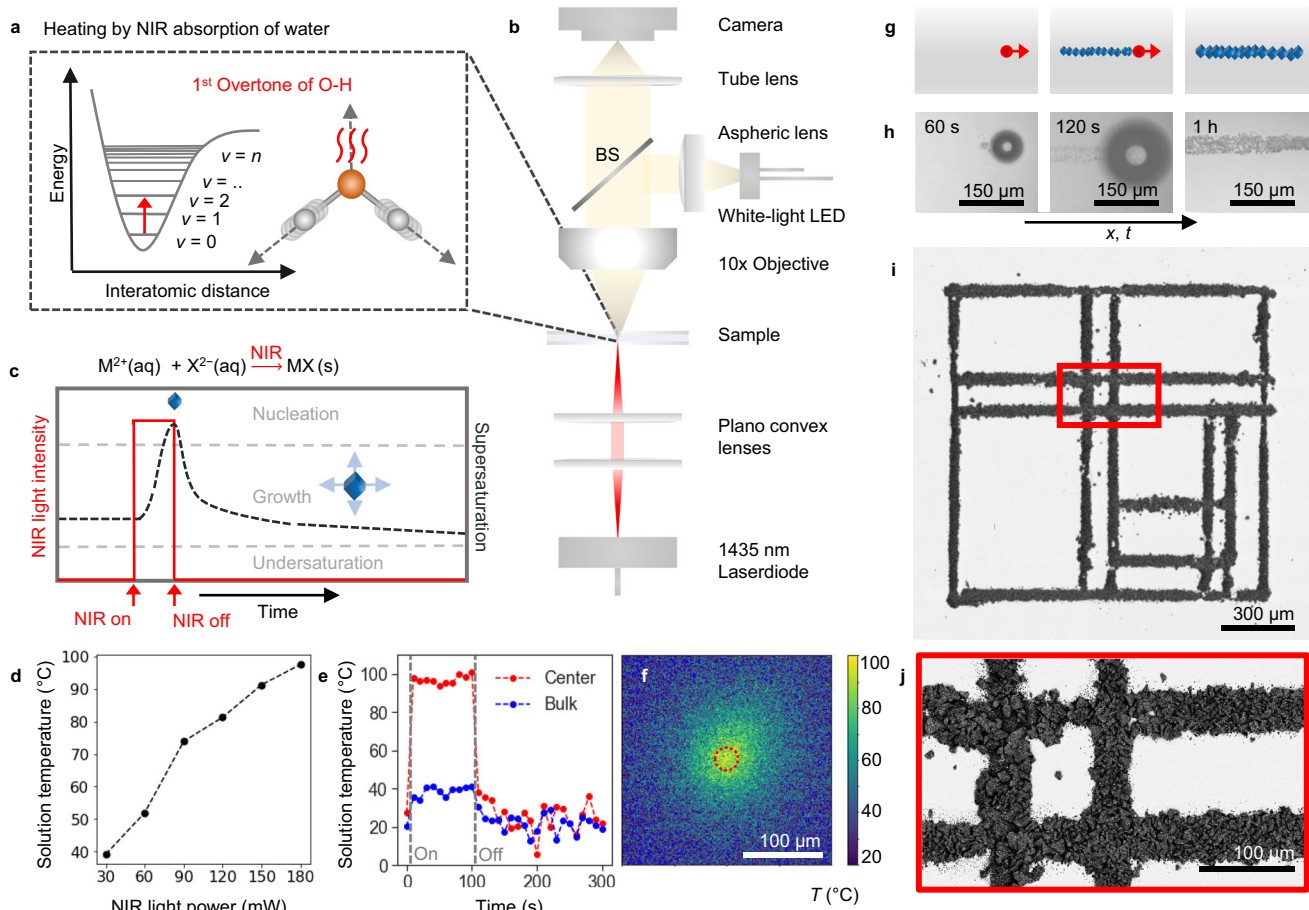

**Fig. 1 | Resonant NIR laser heating for light induced crystallization. a** Light absorption of NIR (1435 nm) by the first overtone of the O-H stretch vibration of water results in local heating. **b** Optical setup in which NIR laser light is focused on the bottom (ITO coated) glass substrate of a reaction cell filled with the precursor solution to locally induce crystallization. Using white light, the process is monitored in-situ. **c** Local heating by NIR light (red line) increases the local supersaturation of salt with retrograde solubility MX (dashed black line) to induce crystallization (blue rhombohedron). **d** Interfacial temperature measurement for a glass substrate, obtained by monitoring the dual-band emission of HPTS, showing the solution temperature in the NIR focal point as function of the incident NIR laser power. **e** Solution temperature upon modulation of the NIR power from 0 to

180 mW NIR irradiation over time for the beam center (red dots) and bulk solution measured 150 µm away from the NIR focal point (blue dots). The NIR was turned on and off manually after 10 and 110 s respectively. **f** Temperature map around the NIR focal point (indicated by red dashed circle) with 180 mW light intensity (pixel size 0.375 µm). **g** Schematic showing that moving the NIR focal spot (red) results in nucleation and growth of a line of crystals with retrograde solubility on an ITO coated glass substrate. **h** In-situ optical microscopy time-lapse showing that moving the NIR laser spot induces local precipitation of CaCO₃. **i** Backscatter SEM image of a NIR laser-written substrate of calcite crystals following a user-defined pattern, with red inset. **j** Showing agglomerates of CaCO₃ organized in intersecting lines on an ITO-coated substrate.

crystallization with spatiotemporal control towards three polymorphs of CaCO$_3$ (vaterite, aragonite, calcite), and other biorelevant carbonate, phosphate and sulfate salts with retrograde solubilities, to demonstrate the generality and sequential control for multimaterial patterning.

## Results and discussion

### Crystallization conditions for NIR laser writing

To control crystal nucleation and growth, we developed a custom-built optical set-up that combines NIR laser light irradiation with in-situ monitoring (Fig. 1b, c). For the NIR laser light irradiation, we focus a fiber-coupled high-intensity laser diode of 1435 nm (spot radius ≈15 μm) on a temperature-controlled and motorized reaction cell holder. The reaction cell is composed of two glass substrates, which are a separated by a viton spacer. The cell is filled with an aqueous precursor solution containing the desired metal carbonate, and the reaction cell is placed in the sample holder. For in situ monitoring, we use a conventional white LED in combination with an objective and CMOS camera opposite from the focal spot.

We quantify the spatial temperature change upon NIR laser irradiation by monitoring the temperature-dependent emission of the photoacid 8-hydroxypyrene-1,3,6-trisulfonate trisodium (HPTS) (see Methods and Supplementary Note 1, Supplementary Fig. 1)[52]. We find that the solution temperature at the focal spot increases approximately linearly as a function of the NIR laser power (Fig. 1d). Moreover, we also observe that the local solution temperature changes within seconds of switching the laser on or off (Fig. 1e), while spatial temperature mapping shows that the heating effect remains localized in the direct vicinity of NIR laser spot (Fig. 1f).

To locally induce nucleation of CaCO$_3$ within the NIR focal spot, we screen the parameters that determine nucleation. We vary in precursor concentrations, additives and surface free energy and monitor how the solutions react on the NIR laser light irradiation (see Supplementary Fig. 2). To prevent nucleation in the bulk solution we increase the nucleation time by adding 100–300 mM NaCl to the precursor solution, which is known to suppress high nucleation rates by reducing the activity coefficients of the precursor ions[53]. Moreover, to promote local nucleation in the laser spot, we use glass slide coated with indium tin oxide (ITO) that absorbs the NIR laser light (see Supplementary Fig. 3) and thereby additionally heats up the substrate precisely in the NIR focal spot. Notably, we observe flow towards the focal spot, which can further support local crystallization by mass transport. Using these insights, we identify an optimum concentration range of calcium chloride salt (CaCl$_2$, 0.5–5 mM), and sodium carbonate salt (Na$_2$CO$_3$, 0.5–2 mM) in a pH-adjusted solution using sodium hydroxide (NaOH, pH 10.7) that gives a metastable precursor solution, in which crystals nucleate only in the NIR irradiated spot.

Based on these findings, we laser-write crystals with a moving NIR focal spot on a substrate (Fig. 1g, h). We prepare a reaction cell of ITO-coated glass substrates containing a precursor solution (1.5 mM CaCl$_2$, 1 mM Na$_2$CO$_3$, 200 mM NaCl, pH 10.7). To induce crystallization, we switch on the NIR laser light (>120 mW) to locally heat up the solution and thereby increase the local supersaturation such that nuclei form in the laser spot. Although not necessary for nucleation, we often observe the formation of gas bubbles in the NIR spot, likely due to local boiling of water[48]. When the NIR spot is moved away from the nuclei, these nuclei continue growing into typical rhombohedron calcite crystal while no new nuclei form outside of the NIR spot, confirming that the precursor solution is supersaturated within the metastable zone for nucleation. Using a velocity of 17.5 μm s$^{-1}$ and a light intensity of 180 mW for the NIR laser light, we write a line of crystals with a width of 30 μm into an abstract microscopic pattern (Fig. 1i, j).

### Light-driven nucleation of single crystals

The NIR laser-induced nucleation offers the opportunity to overcome the stochastic nature of crystallization and form a single crystal of

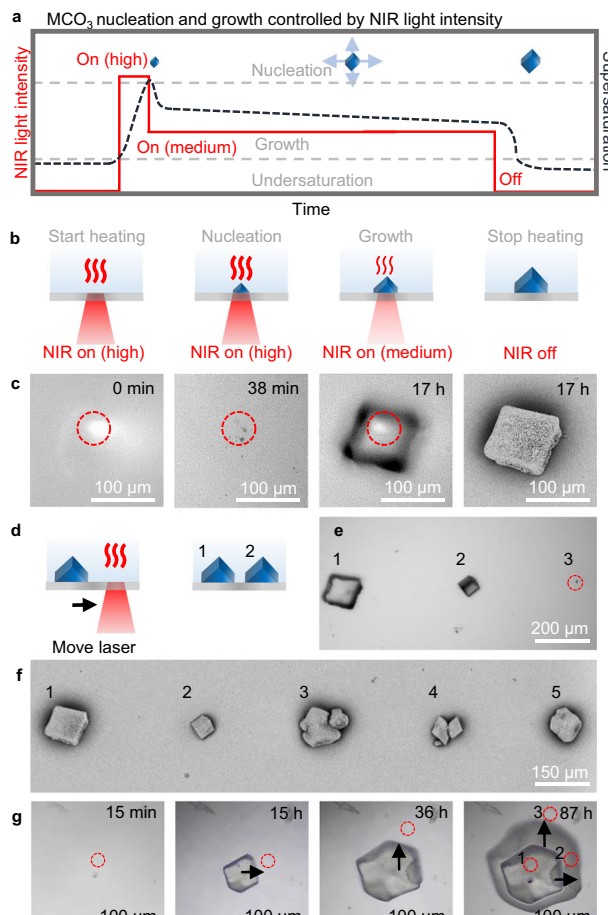

**Fig. 2 | NIR laser-induced nucleation and growth of single calcite crystals with spatiotemporal control. a** Schematic for modulation of the NIR laser light intensity enables separation of nucleation and growth of the retrograde soluble carbonate salts (MCO$_3$). **b** Nucleation of a single calcite crystal by modulating the NIR laser light intensity. High NIR laser light intensity is used to induce nucleation. Lowering the NIR laser light intensity allows growth during NIR laser light irradiation of a single calcite crystal while avoiding new nucleation. **c** In situ optical microscopy time-lapse of a growing single calcite in the NIR focal spot (indicated by red dashed lined circle), and SEM of the final crystal. **d** Successive NIR laser-induced nucleation of calcite with spatial control achieved by moving the NIR focal spot on the substrate. **e** Optical microscope image of successive nucleation of single calcite crystals, with different sizes dependent on their irradiation (see Supplementary Movie 1), indicated by numbers 1–3. **f** SEM image of resulting crystals, indicated by numbers 1–5. **g** Spatiotemporal control over crystal growth of a calcite crystal. After nucleation of calcite (15 min), the NIR laser spot (red dashed circle) is initially moved to the right (15 h) and subsequently to the top (36 h) of the center of the crystal to enhance the growth rate of crystal facets in the direction of the NIR laser spot. Overlay of the timelapse photographs of the asymmetric growing crystal with the positions of the NIR spot at (1) 1 h, (2) 35 h, and (3) 87 h.

CaCO$_3$ in the NIR focal spot (Fig. 2). To achieve this precise resolution, we deliberately lower the nucleation rate by preparing a slightly undersaturated precursor solution (1 mM Na$_2$CO$_3$, 0.5 mM CaCl$_2$, pH 10.7) and equilibrate for two days to suppress undesired spontaneous nucleation. Moreover, to lower the local heating upon irradiation, we replace the ITO-coated glass slides of the reaction cell for uncoated glass slides. Using high NIR laser light intensity (180 mW), we induce nucleation in the focal spot after 30 s (Fig. 2a–c). We then directly reduce the NIR laser light intensity (120 mW) to prevent undesired formation of new nuclei, while still enabling growth of the single crystal. After 17 h, we observe that the nucleus has grown into a calcite crystal in the absence of new nuclei. Turning off the NIR laser stops the

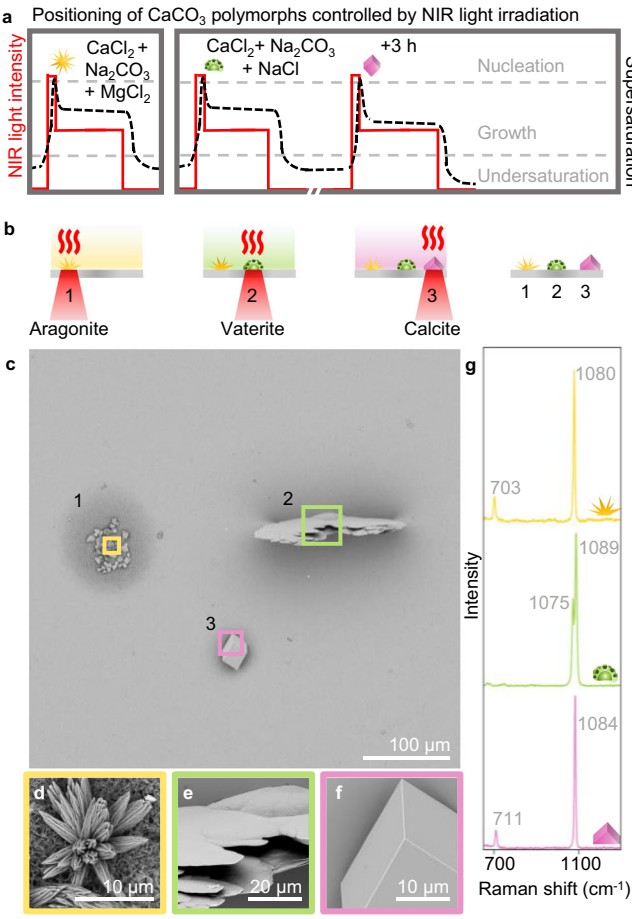

**Fig. 3 | NIR controlled positioning of different CaCO₃ polymorphs on the same substrate. a** Schematic overview of the NIR light schedule (red solid line) and corresponding supersaturation (black dashed line) using NIR irradiation on a CaCO₃ growth solution complemented with MgCl₂ or NaCl at different times. **b** Modification of crystallization conditions induces nucleation and growth of aragonite (yellow, addition of MgCl₂), vaterite (green, addition of NaCl and direct irradiation of freshly prepared precursor solution), and calcite (pink, addition of NaCl and irradiation of 3 h equilibrated precursor solution) with micrometer spacing. **c** SEM image of the three different polymorphs positioned in a triangular pattern. **d**–**f** High magnification SEM images of the three polymorphs. **g** Raman spectra of the three polymorphs, as indicated by the corresponding color lining, yellow (aragonite), green (vaterite), pink (calcite), of the SEM images and schematic growth solutions.

growth but is not followed by dissolution (see Supplementary Fig. 4), indicating that the precursor solution is indeed only slightly undersaturated. Balancing the NIR laser light intensity and solution concentrations thus enables nucleation and growth of a single calcite crystal with spatiotemporal control.

We exploit the spatiotemporal control to position multiple crystals next to each other with micrometer precision. To induce the nucleation and growth of a neighboring CaCO₃ crystal, we move the NIR focal spot 300 μm next to an already grown single calcite crystal (Fig. 2d). Because the precursor solution becomes slightly depleted by precipitation, we observe a trend of longer nucleation times and lower growth rates for later grown crystals. To regain the nucleation and growth rate, we adjust the irradiation time or laser light intensity and sequentially position a line of crystals (Fig. 2e, f; see Supplementary Movie 1), highlighting the level of spatiotemporal control that can be achieved with our NIR pattering strategy.

The spatiotemporal control over the supersaturation suggests that NIR-induced crystallization may be used to modulate the growth

rate of different crystal facets. To explore this potential, we induce the nucleation and growth of a single calcite crystal using NIR irradiation (Fig. 2g). Subsequently, we move the NIR focal spot away from the crystal center to favor the growth in one direction. We observe that all facets of the crystal continue growing, however the facets in the vicinity of the NIR spot grow significantly faster. Hence, NIR can be used to control the growth rate of different crystal facets and thereby steer the crystal morphology.

## Light-driven crystallization of CaCO₃ polymorphs
We test whether NIR laser-induced crystallization is compatible with previously developed methods of polymorph control[54]. Although it is well-known that additives and/or heat both can be used to influence the crystal structure of CaCO₃, it is unclear if NIR heating in combination with additives enables control over both the position and crystal structure. We examine this potential by growing the three CaCO₃ polymorphs calcite, aragonite and vaterite in a triangular pattern (Fig. 3). To enable the growth of aragonite, we use MgCl₂ as additive, which is known to favor the formation of aragonite, to the CaCO₃ precursor solution (2 mM MgCl₂, 1 mM CaCl₂, 1 mM Na₂CO₃, pH 10.7), and load a reaction cell composed of glass substrates (Fig. 3a, b). Indeed, upon irradiation to induce nucleation and growth (5 s 180 mW, 1 h ramping from 80 to 150 mW) we observe needle-like aragonite crystals within the NIR focal spot. Next, we exploit the stability of CaCO₃ to first induce the formation of vaterite and subsequently calcite next to the grown aragonite. We refill the reaction cell with a freshly prepared CaCO₃ precursor solution (300 mM NaCl, 1.5 mM CaCl₂, 1 mM Na₂CO₃, pH 10.7). We enable the nucleation and growth of a single vaterite crystal upon the first NIR laser light irradiation (30 s 180 mW, 30 min, 120 mW respectively). Subsequently, we allow the solution to equilibrate (3 h), after which we reposition the NIR focal spot and induce the nucleation and the growth of a single calcite (2 min 150 mW, 1 h ramping from 90 to 130 mW). Scanning electron microscopy (SEM) imaging and Raman spectroscopy confirm that each crystal is completely formed of the desired polymorph (Fig. 3c–g, Supplementary Fig. 5)[55]. Hence, despite the local temperature gradients we find that additives in the precursor solution can still determine the crystal structure during NIR-induced crystallization, to enable the patterning of different single-crystal polymorphs with micrometer precision.

## Light-driven crystallization of other biorelevant minerals
The principle of resonant NIR laser heating-controlled crystallization can readily be extended to a wide range of other compounds with retrograde solubility. To demonstrate this generality, we position metal carbonate crystals (MCO₃) of barium carbonate (BaCO₃), strontium carbonate (SrCO₃), and CaCO₃—all of which have retrograde solubilities—next to each other on a glass substrate[49,56,57]. Similar to the CaCO₃ precursor solution, we identify the optimum precursor concentration range by in-situ monitoring of the precursor solution upon modulation of the NIR laser light intensity (Fig. 4). Since SrCO₃ and BaCO₃ spontaneously nucleate rather rapidly, we observe that the nucleation occurs with more stochastic fluctuations. Again, we stabilize the precursor solutions by adding NaCl. To crystallize BaCO₃, we fill the reaction cell with the BaCO₃ precursor solution (300 mM NaCl, 0.75 mM BaCl₂, 0.75 mM Na₂CO₃, pH 10.7) and enable nucleation and growth of BaCO₃ upon NIR laser light irradiation (5 min 180 mW, 50 min 180 mW). Subsequently, the reaction cell is flushed with water, and refilled with a SrCO₃ precursor solution (75 mM NaCl, 0.5 mM SrCl₂, 0.25 mM Na₂CO₃, pH 10.1). We move the NIR focal spot 300 μm next to the BaCO₃ crystals and modulate the NIR laser light intensity to induce nucleation and growth of SrCO₃ (2 min 180 mW, 45 min 160 mW). Lastly, the reaction cell is again flushed with water, and refilled with a CaCO₃ precursor solution (300 mM NaCl, 1 mM CaCl₂, 1 mM Na₂CO₃, pH 10.7) to enable nucleation and growth of CaCO₃

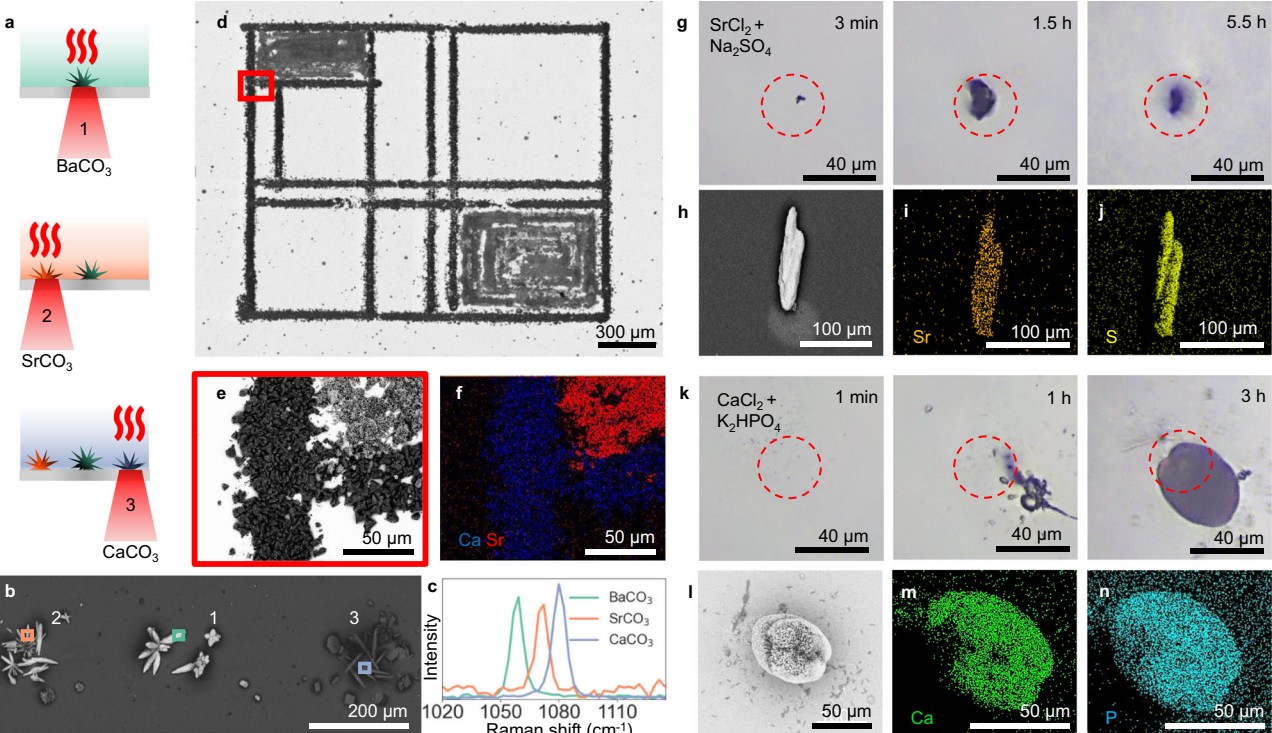

**Fig. 4 | Spatiotemporally controlled nucleation of different retrograde soluble crystals using NIR laser light. a** Schematic for NIR-controlled nucleation of $BaCO_3$ (green), $SrCO_3$ (orange), and $CaCO_3$ (purple). **b** SEM image of the resulting NIR-induced nucleation of $MCO_3$ crystals positioned in a line. **c** Raman spectroscopy of the $MCO_3$ crystals in (**b**), showcasing the counterion-specific shift of the symmetric stretching vibration of carbonate ions. **d** SEM image of NIR laser-induced drawing of $CaCO_3$ and $SrCO_3$, with (**e**) high-resolution inset, and (**f**) EDS map showing calcium (blue) and strontium (red). (**g**) Optical microscopy timelapse series showing the nucleation and growth of strontium sulfate with NIR (dashed red circle). **h** SEM micrograph of the resulting mineral. EDS maps showing (**i**) strontium (orange), and (**j**) sulfur (yellow). **k** Optical microscopy timelapse series showing the nucleation and growth of calcium phosphate upon irradiation with NIR (dashed red circle). **l** SEM micrograph of the resulting mineral. EDS maps showing (**m**) calcium (green), and (**n**) phosphor (blue).

300 μm next to the other side of the $BaCO_3$ crystals (1 min 180 mW, 1.5 h 50 mW). SEM and Raman spectroscopy confirm the positioning of the three different $MCO_3$ crystals with micrometer precision (Fig. 4b, c; Supplementary Fig. 6)[58].

To combine the different levels of NIR laser-induced controlled crystallization, we laser write a pattern of multiple materials. We first fill a reaction cell composed of ITO-coated glass substrates with a meta-stable precursor solution (1.5 mM $CaCl_2$, 1 mM $Na_2CO_3$, 200 mM NaCl, pH 10.7). We laser-write a line pattern of $CaCO_3$ nuclei using 180 mW NIR power and a moving focal spot with a velocity of 17.5 μm s$^{-1}$, and enable the nuclei to grow into crystals for 24 h. To fill in the framework with $SrCO_3$ crystals, we wash the cell twice with water and refill the reaction cell with a $SrCO_3$ precursor solution (200 mM NaCl, 1 mM $SrCl_2$, 0.5 mM $Na_2CO_3$, pH 10.7). Upon irradiation of 180 mW and a moving light beam of 8.5 μm s$^{-1}$, we induce the precipitation of $SrCO_3$ nuclei. Again, we leave the nuclei to grow in the solution for 24 h, and wash the substrate with water and ethanol. SEM with energy dispersion spectroscopy (EDS) mapping shows the hexagonal $SrCO_3$ crystals precisely positioned next to the rhombohedron $CaCO_3$ crystals without any overgrowth (Fig. 4d–f), which emphasizes the high spatial control that can be achieved using NIR laser-induced crystallization of multiple materials.

Besides metal carbonates, many other biorelevant minerals such as metal sulfates and metal phosphates exhibit a retrograde solubility[50,59], hence promising NIR as a more general approach for gaining control over crystallization. We explore this potential by NIR-induced formation of single crystals for two other biorelevant minerals: strontium sulfate and calcium phosphate. For both crystal systems, we modulate the NIR laser power to first induce nucleation at high laser power and subsequently sustain growth of a single crystal while avoiding new nucleation at low laser power. Specifically, for strontium

sulfate we use an aqueous precursor solution containing 1 mM $SrCl_2$ and 1 mM $Na_2SO_4$ and induce nucleation using 3 min irradiation with 180 mW NIR followed by 5.5 h with 130–150 mW to sustain growth (Fig. 4g). For calcium phosphate, we use an aqueous precursor solution of 1 mM $CaCl_2$ and 1 mM $K_2HPO_4$ and 300 mM NaCl to prevent formation of amorphous precipitate and induce nucleation using irradiation of 1 min with 180 mW, followed by growth at 3 h with 130–180 mW to sustain growth (Fig. 4k). Raman spectra are in good agreement with the biomineral celestine for strontium sulfate and the bone mineral hydroxyapatite for calcium phosphate. SEM and EDS confirm the formation of single crystals of strontium sulfate and calcium phosphate. (Fig. 4h, i, l–n, and Supplementary Fig. 7)[60,61], hence demonstrating the versatility and generality of NIR for inducing crystallization of retrograde minerals with spatiotemporal control.

We here introduce resonant NIR laser heating for gaining precise user-defined spatiotemporal control over crystal nucleation and growth of compounds with retrograde solubilities. To demonstrate the proof-of-concept, we tailor precursor solutions and modulation of NIR laser light intensities to pattern various biorelevant retrograde soluble crystals with micrometer precision applied on both chemical composition and polymorphism.

The simplicity and versatility of our strategy makes NIR-induced crystallization accessible and implementable in widely used crystallization techniques. Importantly, control over the concentrations of the precursor solution and dynamic modulation of the light intensity provides a large parameter space for avoiding undesired spontaneous nucleation while also gaining spatiotemporal control over both nucleation as well as growth of crystals. In fact, we foresee that exploiting well-established crystallization techniques such as microfluidics, diffusion-limiting gels or topographic confinements can even

further improve the resolution. Additionally, higher laser powers and narrow focusing may further improve the resolution of the local heating. Furthermore, tuning the NIR wavelength enables resonant heating for a wide range of solvents, which extends the application potential of NIR-induced crystallization towards a broad class of desirable functional materials with retrograde solubility. For example, in case of perovskites, electrical heating already showed promising results for growing single crystals[51], but the precise spatiotemporal control of NIR-induced crystallization can now be exploited to grow these widely studied semiconductors with exact control over shape and position. We foresee that NIR-induced crystallization may also be suitable for quantitatively controlling the local supersaturation and studying the role of local heating on crystallization dynamics and chemical reactions with spatiotemporal control. Furthermore, the narrow chemoselective absorption wavelength of NIR may enable synergistic combinations with orthogonal light-induced (photo-chemical) reactions, akin to double photochemical activation, for control over self-assembly in three dimensions[62].

The principles presented here can readily be applied to a broad class of crystallization and self-assembling systems. Specifically, bio-minerals such as the bone component calcium phosphate and most common biomineral calcium carbonate have retrograde solubilities. In addition, NIR light is well-suited for bio-active applications due to its mild energy as compared to UV light[47], thus offering the opportunity to manipulate mineralization processes in vivo. In artificial systems, we foresee that our optical setup for NIR heating can directly be used to exploit temperature-dependent phase separations for light-directed assembly of nanoparticles, e.g. temperature sensitive assembling, into user-defined structures[63]. In addition, the metal carbonate salts that we use for the proof-of-concept can readily undergo ion-exchange reactions to yield a wide range of shape-controlled materials with desirable electronic, optical, and chemical properties such as semiconductors, metals, and catalysts[64,65]. By controlling the patterning of nucleation and growth, our work can also contribute and push boundaries in the area of hierarchical mineralization, which continues to be a major challenge in materials science. For instance, the precipitation of car-bonate salts and silica may be directed using NIR-defined light patterns to create highly complex microscopic three-dimensional nano-composites according to exact designs[66,67]. This creates possibilities for steering the assembly of functional components such as optical components for directional emission and synthetic materials for regenerative medicine and dentistry[68,69], hence moving towards the magnificent micro- and nanoscopic control that organisms exert to form biominerals with exquisite performance[70]. In conclusion, NIR-induced heating offers opportunities for light-controlled bottom-up manufactory of highly functional materials with user-defined control over position, size, shape, and crystal structure.

## Methods

### Preparation of precursor solutions

Stock solutions are prepared by dissolving the desired compound in degassed water: calcium chloride dihydrate ($CaCl_2 \cdot 2H_2O$) (Sigma-Aldrich, ACS reagent, ≥99%) (28 mg, 19 mL, 10 mM), sodium chloride (NaCl) (Merck, ACS reagent, ≥99.0%) (2340 mg, 80 mL, 500 mM), barium chloride dihydrate ($BaCl_2 \cdot 2H_2O$) (VWR, ≥99.0%) (25 mg, 10 mL, 10 mM), strontium chloride hexahydrate ($SrCl_2 \cdot 6H_2O$) (Sigma-Aldrich, ACS reagent, ≥99.0–102.0%) (27 mg, 10 mL, 10 mM), magnesium chloride hexahydrate ($MgCl_2 \cdot 6H_2O$) (Sigma-Aldrich, ACS reagent, ≥99.0–102.0%) (20 mg, 10 mL, 10 mM), sodium sulfate ($Na_2SO_4$) (Merck, ACS reagent, ≥99.0%) (14 mg, 10 mL, 10 mM), potassium phosphate dibasic ($K_2HPO_4$) (VWR, ACS reagent, ≥98%) (17 mg, 10 mL, 10 mM), and NaOH (VWR, pellets ACS, ≥97.0%) (4 mg, 10 mL, 10 mM). 10 mM Sodium carbonate ($Na_2CO_3$) (Sigma-Aldrich, ACS reagent, ≥99.5%) stock solution is prepared by dissolving 21 mg in 20 mL degassed water adjusted with NaOH stock solution to pH 10.7.

Typically, the precursor solution is prepared by freshly mixing 0–1.8 mL 500 mM NaCl stock solution with 150–1500 μL 10 mM $MCl_2$ stock solution, which then is added to 150–600 μL 10 mM $Na_2CO_3$, $Na_2SO_4$ or $K_2HPO_4$ stock solution. Then the pH is adjusted with 0–150 μL 10 mM NaOH stock solution and the solution is diluted with 0.15–2.55 mL degassed water to have a final total volume of 3 mL. The additive solution ($MgCl_2$) is first diluted with degassed water before added to the $MCl_2$ solution, which is then mixed with the $Na_2CO_3$ and NaOH solution to obtain a final total volume of 3 mL. After an equili-bration time (0–48 h), the final growth solution is injected in a closed reaction cell, made of two (ITO-coated) glass substrates that were first ozone cleaned (10 min) and a viton spacer (2–3 mm spacing, 1.3–1.9 mL volume). After NIR-induced crystallization, the substrates are dis-assembled in degassed water and carefully washed with ethanol.

### Custom-built NIR-irradiation microscopy

In the custom-built setup used for resonant infrared laser heating, three main parts can be identified: the NIR laser diode irradiation part, the reaction cell holder stage, and the imaging part. In the irradiation part, the output light of a fiber-coupled 1435 nm laser diode (Alcatel A1948FBG, 180 mW output power) is collimated by a plano-convex lens (Thorlabs LA1422, focal length ($f$) = 4.0 mm). After the collimation, the beam is refocused with another plano-convex lens (Thorlabs LA1422, $f$ = 4.0 mm) on the sample. We control the light intensity by adjusting the driving current of the laser diode. The sample holder stage consists of a custom-made temperature cell on top of a motor-ized translation table. The solution temperature is controlled by a bath and circulation thermostat (Huber CC-K6). Motion control is provided by piezo inertia actuators (Thorlabs PIAK10) that have a typical step size of 20 nm. In the imaging part, the light of a mounted cold white light LED (Thorlabs MCWHL5) is collected by an aspheric lens (Thor-labs ACL2520U-DG6-A, $f$ = 20.0 mm). A 10:90 beamsplitter (Thorlabs BSN10R) directs the light through a 10x/0.30 magnification objective (Nikon Plan Fluor) to the sample. The reflected images are collected with the same objective lens and transmitted through the beams-plitter. With the help of a tube lens (Thorlabs AC254-200-A-ML), ima-ges are recorded by a CMOS camera (Basler Ace, acA1920-40gc).

### Raman spectroscopy

Raman spectroscopy measurements are carried out using a confocal Raman microscope (Witec, Alpha 300R) with an EMCCD detector (Andor, Newton, DU970P-BVF-355). Excitation is performed with a 532 nm laser while spectra are collected in a spectrometer, equipped with a diffraction grating with a groove density of 600 grooves $mm^{-1}$. The spectra are each 20 times averaged with an integration time of 1 s.

### Scanning electron microscopy

SEM imaging is performed using an FEI Verios 460.

### Temperature measurements

For monitoring spatially-resolved interfacial temperature changes upon NIR laser irradiation, we developed a fluorescence-based experimental method that utilizes the two temperature-dependent emission bands of the photoacid 8-hydroxypyrene-1,3,6-trisulfonic trisodium salt (HPTS) (Sigma-Aldrich, ≥96%) that overlap with the green (G) and blue (B) channels of the CMOS sensor in our transmis-sion microscope, allowing us to monitor their ratios without using a spectrometer. 5 mM HPTS solutions are prepared by dissolving 14 mg HTPS in 2.5 mL degassed water and 2.5 mL 1 M perchloric acid (Sigma-Aldrich, ACS reagent, 70%). In our setup, we illuminate the solution with a low-intensity 400 nm LED from the same side as the NIR laser diode. We record the resulting fluorescence images with an RGB camera. To evaluate local temperature changes induced by the NIR laser heating, we construct a calibration curve, based on G:B intensity ratios recorded at different NIR heating power levels. To this aim, we

heat the HPTS solution in the reaction cell with a bath and circulation thermostat (Huber CC-K6) to 83 °C under the illumination of the 400 nm LED and record the RGB channels. Measurements are repeated with the same procedure for a reaction cell build of ITO-coated substrates. To measure the bulk temperature upon long NIR irradiation, a thermocouple is placed ca. 1 mm from the focal spot.

## UV/vis/NIR absorption of glass and ITO
Steady-state external transmission measurements are carried out using a LAMBDA 750 UV/vis/NIR spectrophotometer (Perkin Elmer), equipped with a deuterium and tungsten excitation source, an InGaAs detector and an integrating sphere. The absorbance measurements are done for the range of 1000–1600 nm.

## Reporting summary
Further information on research design is available in the Nature Portfolio Reporting Summary linked to this article.

## Data availability
All data and coding supporting the findings of this study are available from the corresponding author upon request.

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

## Acknowledgements

The authors thank Prof. Dr. Albert M. Brouwer for fruitful discussions, Dr. Verena Neder for assisting with the first exploratory experiments, and Simon Lepinay for Raman spectroscopy measurements. This work is part of the Vernieuwingsimpuls Vidi research program "Shaping up materials" with project number 016.Vidi.189.083, which is partly financed by the Dutch Research Council (NWO).

## Author contributions

The idea, design and concept of the research were developed by M.H.B. and W.L.N. The methodology was developed by M.H.B. and W.L.N. The experimental set-up was developed by M.H.B., B.A., M.K., and H.S. The experiments were performed by M.H.B. The data was analyzed by M.H.B., B.A., and W.L.N. The project and research were managed and supervised by W.L.N. The manuscript was written and edited by M.H.B. and W.L.N.

## Competing interests

The authors declare no competing interests.
