## [Peer Review File · Nature Communications]

Light-driven nucleation, growth, and patterning of biorelevant crystals using resonant near-infrared laser heatingReviewers' Comments:

Reviewer #1:

Remarks to the Author:

The authors describe a elegant procedure to produce crystal formation by local heating of solution of compounds with retrograde solubility where NIR light is focussed to create supersaturation. The technique is further demonstrated to induce different polymorphs of CaCO₃ by tuning the solution conditions. The authors further illustrate that complex shapes can be laser-written by automating the laser motion along the glass slide.

I recommend publication of the manuscript after major concerns mentioned below are addressed.

1) What is the time dependent temperature in and around the laser spot ? This is a critical experimental parameter and essential for wide spread applicability of the method. The only indication of temperature is a reference to a different paper in main text "NIR laser light has been shown to locally heat up water with 50 °C in a 10 micrometer diameter light spot within milliseconds, suggesting great potential for inducing crystallization of retrograde soluble minerals with precise spatiotemporal control without any need of additives." No quantification of the temperature in the presented experimental setup has been provided for this setup. This should be improved. None of the figures only show temperature and local supersaturation quantitatively. The authors provide only qualitative hand drawn lines in Figures 1 &2 no absolute numbers.

2) How does the local temperature and supersaturation change in and around the laser spot after the first nuclei form?

3) How does the temperature and supersaturation change as the laser is shut down. The authors illustrate that the supersaturation decreases after the laser is shut down in Figure 1c. How fast does this decrease occur?

4) The authors do not explain why a bubble observed in Figure 1c form. Why does it form?

5) Does the bubble form in each experiment? Does it alter local supersaturation and temperature? Does it happen at only high intensities. More effort has to be placed to described the bubble formation. I see the explanations as very vague.

6) The single crystal growth of CaCO₃ is reported as 17 hours. This is rather long. Can this time be tuned?

7) Can different crystal be grown on to each other? say starting with CaCo₃ then a different polymorph on the top.

Reviewer #2:

Remarks to the Author:

The manuscript addresses spatiotemporal control of crystal nucleation and growth in solution, which is – as the authors point out – very difficult to realize. The idea of using a NIR laser to control local temperature in the solution is an original and very promising approach. However, several fundamental aspects are left open or, at least, are not properly addressed in the manuscript.

1) Spatial control of precipitation has already been achieved by various methods, for example using self-assembled monolayers that control nucleation. What would really make a difference is the simultaneous control of spatial and temporal nucleation and growth (as sketched in Fig. 2). While local heating is promising, the authors do not consider the fact that spatial and temporal resolution are in

fact coupled with this technique. If we just take $D=0.15 \text{ mm}^2/\text{s}$ as the thermal diffusivity in water, then the typical diffusion distance $\text{Sqrt}(D t)$ is in the order of 10 cm for the 17 hour treatment mentioned in Fig. 2. Therefore, I would expect the whole bath to slowly heat up and, thereby, change the conditions even far away from the spot hit by the NIR laser. I believe that it is essential to study and report the magnitude of this temperature coupling by measuring temperature fields in the bath. This should also allow the estimation of a spatial resolution of the process that should in fact depend on the laser treatment time.

2) Spatiotemporal coupling is not a major issue, when just one crystal is nucleated and grown as shown in Fig. 2 (but this can also be achieved by traditional methods). The challenge is really to understand what happens to other crystals in the vicinity when the laser treatment needs hours to grow one crystal.

3) Another point that needs addressing is the fundamental difference between nucleation and growth with respect to this spatiotemporal coupling. Nucleation happens beyond a critical supersaturation and – translated into temperature – beyond a sharply defined temperature above which supersaturation is sufficient for nucleation. This means that an increase of the bath temperature away from the laser spot will not induce nucleation as long as it stays sufficiently small. This is probably why it was possible to draw the mineralized lines in Fig. 1, which are obviously made by a large number of small crystallites, probably due to multiple nucleation events. A significant crystal growth is not seen in this picture (as opposed to Fig. 2). The situation is, indeed, totally different for crystal growth (or dissolution) once they are nucleated: These processes are temperature-dependent, but they do not stop or start at a well-defined critical value (although there might be a switch from growth to dissolution). For this reason, I would expect crystal growth to be much more difficult to control in a spatiotemporal manner. Therefore: can spatiotemporal control of crystal growth actually be demonstrated?

4) Fig. 3 is interesting in this respect, but it does not show a proper spatiotemporal control of crystal growth, since the solution is modified over time.

In summary, I think that this is very promising work, but that the claims about spatiotemporal control of crystal nucleation and growth have not been demonstrated at this point. Perhaps, it turns out that only nucleation can be controlled in this way. This would be very interesting in itself, but then the claims in the manuscript should be adapted. Some additional information on temperature fields (as described above) would also be needed.

Reviewer #3:

Remarks to the Author:

I previously reviewed this article for Nature Chemistry, where I felt that the work was elegant and achieved spatio-temporal control with a better spatial resolution than had been previously demonstrated. However, I did not find it of sufficient significance for publication in that journal, and felt that the polymorph control achieved was somewhat over-sold.

I think that the authors have taken all of the reviewers' comments into account and have definitely improved their article. In particular, the description of polymorph control is now well-balanced. However, I still remain rather unconvinced of the generality/ overall importance of the approach. It is notable that the quality of the results are much higher for calcium carbonate than any of the other systems investigated. I am therefore not sure that the article merits publication in Nature Comms. The idea itself (using light to pattern crystallisation) is not novel, although the authors have developed a new strategy that delivers much better results than previously reported. The control achieved over CaCO_3 is excellent, but the other systems are much less impressive.

I have just a few other comments:

1. What is the relative importance of the heating of the water and the heating of the ITO? How did the data compare for the calcite experiments if glass was used rather than ITO? Why was glass rather than ITO used in the experiments generating vaterite and aragonite?
2. In my previous review I queried why some reaction solutions were equilibrated for hours prior to initiating crystallisation. Why was this? What was the significance of the length of the equilibration time?
3. I can't find the length of the scale bar in Fig 2C.
4. P13 "for light-controlled bottom-up manufacture of highly functional materials..."

Reviewer #4:

Remarks to the Author:

The paper by Bistervels et al reports on the use of NIR-based heating to control the nucleation and growth of MCO_3 crystals with spatio-temporal control. This is an important and timely goal and the method proposed is novel. However, this novelty should be better presented and some additional discussion around its rationale and implications would help the manuscript. In general, the manuscript is well-written and presented clearly.

In my opinion the manuscript is suitable for publications but should address first the following points.

The rationale behind controlling mineralization spatially and temporally could be clearer from the beginning. The first sentence of the introduction suggests this importance but it would help to elaborate and give specific examples...for example the resulting hierarchy dental enamel allows XXX, the organization in nacre allows YYY, etc. Also, in this paragraph, it would be good to also incorporate and mentioned studies using topographies as a technique to guide mineralization.

Figure 1B should provide more detail. The experimental setup is an important contribution of the study so the setup should be presented in more detail.

Another point to improve relates to the underlying mechanism of single crystal nucleation and growth for the different polymorphs, particularly explaining the role of fluctuating laser power in controlling nucleation and crystal growth processes. Here, a figure describing this underlying mechanism of mineralization/patterning at the molecular/nano scale would strengthen the manuscript.

Page 5, the use of additives is mentioned but it is not clear what additives, their concentrations, and their role in MCO_3 crystallization. Please elaborate.

Page 6, authors mentioned that the use of a high-power laser (180 mW) to induce nucleation and later reduced laser power (120 mW) to precisely grow single crystal and to avoid new nucleation. It is not clear why high-power laser won't create multiple nuclei in the first instance during nucleation. Please discuss. Is there a threshold of laser power for preventing new nucleation?

Localized heating using electricity has been demonstrated to generate single crystals perovskite in the past (10.1021/acs.cgd.2c00833). The authors should discuss similar studies and directly differentiate themselves. It is not clear how the authors monitored this localized temperature change upon NIR exposure? What was the set up used for monitoring? This should at least be mentioned in the methods section.

Also, localized heating due to vibrational absorption of water in the NIR region has been previously exploited in different applications (10.1021/nn500702g). The discussion should also mention these kinds of studies and highlight the novelty of the current work (control of mineralization, etc).

By controlling the patterning of nucleation and growth, the work also contributes and pushes boundaries in the area of hierarchical mineralization, which continues to be a major challenge in materials science. This is part of the novelty of the study and would be worth discussing by comparing to state-of-the-art (10.1038/s41467-018-07658-0, 10.1126/science.aah6350, etc) particularly around the micro-to-macro scale control and connecting back to idea of organisms in nature...eg 10.1002/adhm.201800178, etc.

The patterning reported is novel and an important advancement. However, the study and manuscript would be significantly strengthened if the versatility of the technique is demonstrated for example using at least one of the systems mentioned on page 13 (calcium phosphate, nanoparticles assembly, and/or perovskites).

In the discussion or conclusion there should be some mentioning of potential applications of the patterned MCO₃ crystals.

Please revise minor consistency errors such as scale bar format (Fig 2C), line on top of page 8, etc.

Reviewer rebuttal

NCOMMS-23-09486 with former title “Light-controlling nucleation, growth and patterning of metal carbonate crystals using resonant near-infrared laser heating”, and new title “Light-controlling nucleation, growth and patterning of biorelevant crystals using resonant near-infrared laser heating”

We would like to thank the reviewers for their time and detailed and constructive comments, which we address point-by-point below. The reviewers’ comments are pasted below in italics while our response is in regular font. Changes in the manuscript are highlighted in yellow.

Reviewer #1:

The authors describe a elegant procedure to produce crystal formation by local heating of solution of compounds with retrograde solubility where NIR light is focussed to create supersaturation. The technique is further demonstrated to induce different polymorphs of CaCO₃ by tuning the solution conditions. The authors further illustrate that complex shapes can be laser-written by automating the laser motion along the glass slide.

I recommend publication of the manuscript after major concerns mentioned below are addressed.

Comment: *What is the time dependent temperature in and around the laser spot? This is a critical experimental parameter and essential for wide spread applicability of the method. The only indication of temperature is a reference to a different paper in main text "NIR laser light has been shown to locally heat up water with 50 °C in a 10 micrometer diameter light spot within milliseconds, suggesting great potential for inducing crystallization of retrograde soluble minerals with precise spatiotemporal control without any need of additives." No quantification of the temperature in the presented experimental setup has been provided for this setup. This should be improved. None of the figures only show temperature and local supersaturation quantitatively. The authors provide only qualitative hand drawn lines in Figures 1 & 2 no absolute numbers.*

Response: We thank the reviewer for the suggestions. To address the reviewer’s comment and reviewer #2, we developed a new, fluorescence-based method to quantify the local interfacial heating induced by the incident NIR laser. In this method, we monitor the *ratio* of the two emission bands of acidified solutions (0.5 M HClO₄ added) of the photoacid 8-hydroxypyrene-1,3,6-trisulfonate trisodium salt (HPTS, 5 mM) at different temperatures. We excite this molecule with a commercial UV LED centered at 400 nm. Using this method, we find that the temperature in the irradiated spot increases approximately linear with incident NIR laser power, reaching ca. 90 °C at 180 mW (Figure R1).

The mapping of the heat profile shows that the heating is localized around a radius of ca. 50 μm for a NIR spot with a radius of ca. 15 μm (Figure R1B). Moreover, heating and cooling in the NIR laser spot happens within seconds of switching the laser on and off (Figure R1C). We added these new quantitative data to the main text and Figure 1, and details on the experiments in the Methods section and Supporting Information.

Figure R1. Local temperature measurements using the temperature dependent dual-band emission of HPTS. (A) Solution temperature in the NIR focal point as function of the NIR laser power; (B) Temperature map around the NIR focal point (indicated by black dashed circle) with 180 mW light intensity. (C) Solution temperature upon 180 mW NIR irradiation over time for the beam center (red dots) and bulk solution 150 μm from the NIR focal point (blue dots). The NIR was manually turned on and off after 10 and 110 seconds respectively.

Comment: *How does the local temperature and supersaturation change in and around the laser spot after the first nuclei form?*

Response: We find that the local temperature in the irradiated area changes within seconds after switching the NIR laser on and off, while the temperature of the bulk solution (150 μm measured from the NIR focal center) remains virtually unchanged (Figure R1C). Since these compounds have a retrograde solubility, consequently the supersaturation goes up. We clarified the role of the local supersaturation and added the measurements on the local temperature in the main text with further details in the Methods and SI.

Comment: *How does the temperature and supersaturation change as the laser is shut down. The authors illustrate that the supersaturation decreases after the laser is shut down in Figure 1c. How fast does this decrease occur?*

Response: Using the new temperature measurements, we find that the solution cooled down to the initial temperature within seconds (Figure R1C). Since the supersaturation is proportional to the temperature, the supersaturation also drops within seconds. We adjusted the main text and added these data to the SI to clarify the temperature change.

Comment: *The authors do not explain why a bubble observed in Figure 1c form. Why does it form?*

Response: The bubbles likely form due to boiling of water. We clarify this in the new version of the main text.

Comment: *Does the bubble form in each experiment? Does it alter local supersaturation and temperature? Does it happen at only high intensities. More effort has to be placed to described the bubble formation. I see the explanations as very vague.*

Response: The bubbles only form when the solution temperature approaches the boiling point of water. Importantly, we find that the formation of bubbles is not essential to induce crystallization. We now emphasize this in the main text.

Comment: *The single crystal growth of CaCO₃ is reported as 17 hours. This is rather long. Can this time be tuned?*

Response: Yes, the growth time can be tuned. In the experiment shown in Figure 2 we grew a single crystal in 17 hours. In this experiment, we deliberately selected a rather low initial precursor concentration to avoid the undesired nucleation of multiple crystals. However, we can either increase the precursor concentration or increase the light intensity to increase the

temperature dependent supersaturation and thereby shorten the growth time. We now emphasize this tunability in the conclusion section of the main text.

Comment: Can different crystal be grown on to each other? say starting with CaCO_3 then a different polymorph on the top.

Response: We thank the reviewer for this interesting suggestion. To preliminary test this option, we induced the nucleation and the growth of a few large calcite crystals upon NIR irradiation (1.5 mM CaCl_2 , 1 mM Na_2CO_3 , 300 mM NaCl, pH 10.7, Figure R2A). Subsequently, we replaced the growth solution in the reaction cell with a growth solution favoring aragonite (0.5 mM CaCl_2 , 0.5 mM Na_2CO_3 , 300 mM NaCl, 1 mM MgCl_2 , pH 10.7). By placing the NIR focal spot at the already grown calcite we aimed to induce the growth of aragonite on top of the calcite upon irradiation. We indeed observed the growth of aragonite on top of the calcite crystal (Figure R2B-D). Thus, the NIR irradiation can be used to grow different polymorphs on top of each other. We chose to not include these results in this manuscript, as further experiments may be required to explore the potential of this.

A 1.5 mM CaCl_2 , 1 mM Na_2CO_3 , 300 mM NaCl, pH 10.7

B 0.5 mM CaCl_2 , 0.5 mM Na_2CO_3 , 300 mM NaCl, 1 mM MgCl_2 , pH 11

Figure R2. NIR induced crystallization of CaCO_3 polymorphs on top of each other. **(A)** Timelapse of the nucleation and growth of calcite crystals using a NaCl complemented growth solution (1.5 mM CaCl_2 , 1 mM Na_2CO_3 , 300 mM NaCl, pH 10.7) during 46 hours irradiation of a moving NIR focal spot with a varying light intensity. **(B)** Timelapse of the nucleation and growth of aragonite crystals positioned on top of the calcite crystals using a MgCl_2 and NaCl complemented growth solution and 4.5 h NIR irradiation with varying light intensity, the NIR focal spot is indicated using a red dashed circle. **(C)** SEM micrograph **(D)** and zoom-in, indicating that calcite was overgrown with aragonite.

Reviewer #2:

The manuscript addresses spatiotemporal control of crystal nucleation and growth in solution, which is – as the authors point out – very difficult to realize. The idea of using a NIR laser to control local temperature in the solution is an original and very promising approach. However,

several fundamental aspects are left open or, at least, are not properly addressed in the manuscript.

In summary, I think that this is very promising work, but that the claims about spatiotemporal control of crystal nucleation and growth have not been demonstrated at this point. Perhaps, it turns out that only nucleation can be controlled in this way. This would be very interesting in itself, but then the claims in the manuscript should be adapted. Some additional information on temperature fields (as described above) would also be needed.

Comment: Spatial control of precipitation has already been achieved by various methods, for example using self-assembled monolayers that control nucleation. What would really make a difference is the simultaneous control of spatial and temporal nucleation and growth (as sketched in Fig. 2). While local heating is promising, the authors do not consider the fact that spatial and temporal resolution are in fact coupled with this technique. If we just take $D=0.15$ mm²/s as the thermal diffusivity in water, then the typical diffusion distance $\text{Sqrt}(D t)$ is in the order of 10 cm for the 17 hour treatment mentioned in Fig. 2. Therefore, I would expect the whole bath to slowly heat up and, thereby, change the conditions even far away from the spot hit by the NIR laser. I believe that it is essential to study and report the magnitude of this temperature coupling by measuring temperature fields in the bath. This should also allow the estimation of a spatial resolution of the process that should in fact depend on the laser treatment time.

Response: We like to thank the reviewer for emphasizing this point. Based on the comment of this reviewer and reviewer #1, we quantified the local heating as function of NIR power using laser induced fluorescence (LIF). Using the dual-band temperature dependent emission of 8-hydroxypyrene-1,3,6-trisulfonate trisodium salt (HPTS), we find that the temperature in the irradiated spot increases approximately linear as function of the NIR laser power, reaching ca. 90 °C at 180 mW (Figure R3). Mapping of the heat profile shows that the heating is very local (Figure R3B). Moreover, heating and cooling are almost instantaneously, within seconds the solution heats up and cools down in the NIR light spot when switching the laser on and off respectively (Figure R3C). Also during long term irradiation we do not observe a notable temperature change in the bulk solution. We added these new quantitative data to the main text, and details on the experiments in the Methods section and Supporting Information.

Figure R3. Local temperature measurements using the temperature dependent emission of HPTS. (A) Temperature measurement using the two banded emission of HPTS: (i) Solution temperature in the NIR focal point as function of the NIR laser power; (ii) Temperature map around the NIR focal point (indicated by black dashed circle) with 180 mW light intensity. (C) Solution temperature upon 180 mW NIR irradiation over time for the beam center (red dots) and bulk solution 150 µm from the NIR focal point (blue dots). The NIR was turned on and off manually after 10 and 110 seconds respectively.

Comment: Spatiotemporal coupling is not a major issue, when just one crystal is nucleated and grown as shown in Fig. 2 (but this can also be achieved by traditional methods). The challenge

is really to understand what happens to other crystals in the vicinity when the laser treatment needs hours to grow one crystal.

Response: Using the laser induced fluorescence of HPTS, we mapped the solution temperature in and around the NIR spot with a laser power of 180 mW, showing that heating is very local and the temperature of the bulk solution hardly changes over time (Figure R3). These new measurements are consistent with our previous experiments in which we found no observable influence on the crystallization of crystals that have already formed when they come in the vicinity of the NIR laser spot. We clarified this spatiotemporal control now in the main text and added the new experiments on the temperature measurement in the main text, Methods and SI.

Comment: Another point that needs addressing is the fundamental difference between nucleation and growth with respect to this spatiotemporal coupling. Nucleation happens beyond a critical supersaturation and – translated into temperature – beyond a sharply defined temperature above which supersaturation is sufficient for nucleation. This means that an increase of the bath temperature away from the laser spot will not induce nucleation as long as it stays sufficiently small. This is probably why it was possible to draw the mineralized lines in Fig. 1, which are obviously made by a large number of small crystallites, probably due to multiple nucleation events. A significant crystal growth is not seen in this picture (as opposed to Fig. 2). The situation is, indeed, totally different for crystal growth (or dissolution) once they are nucleated: These processes are temperature-dependent, but they do not stop or start at a well-defined critical value (although there might be a switch from growth to dissolution). For this reason, I would expect crystal growth to be much more difficult to control in a spatiotemporal manner. Therefore: can spatiotemporal control of crystal growth actually be demonstrated?

Response: We agree with the reviewer's reasoning. To further demonstrate the level of spatiotemporal control that can be achieved by NIR induced crystallization, we performed a new experiment in which we directed the growth of a crystal. For this we nucleated a single calcite crystal and subsequently moved the NIR laser away from middle of the crystal towards the right side, followed by the top side of the crystal to promote growth on these sides of the crystal (Figure R4). We find that the irradiated side of the crystal grows faster, demonstrating that spatiotemporal control over growth is indeed possible. We added a new paragraph on this experiment in the main text and added the results in Figure 2G.

Figure R4. Spatiotemporal control over crystal growth of a calcite crystal. After nucleation of calcite (i), the NIR laser spot (red circle) is initially moved to the right (ii) and subsequently to the top (iii) of the center of the crystal to promote growth in this direction. (iv) The overlay of the timelapse photographs of the asymmetric growing crystal with the positions of the NIR spot at (1) 1 hour, (2) 35 hours, and (3) 87 hours.

Comment: Fig. 3 is interesting in this respect, but it does not show a proper spatiotemporal control of crystal growth, since the solution is modified over time.

Response: In Fig. 3, we demonstrate control over the moment and place of crystallization. For this spatiotemporal control, indeed the solution is changed, but it is also essential that we select

the right light intensity, irradiation time, and position of the NIR focal spot, such that we determine the position, growth rate, and crystal size of the three different polymorphs of CaCO_3 .

Reviewer #3:

I previously reviewed this article for Nature Chemistry, where I felt that the work was elegant and achieved spatio-temporal control with a better spatial resolution than had been previously demonstrated. However, I did not find it of sufficient significance for publication in that journal, and felt that the polymorph control achieved was somewhat over-sold. I think that the authors have taken all of the reviewers' comments into account and have definitely improved their article. In particular, the description of polymorph control is now well-balanced. However, I still remain rather unconvinced of the generality/ overall importance of the approach.

It is notable that the quality of the results are much higher for calcium carbonate than any of the other systems investigated. I am therefore not sure that the article merits publication in Nature Comms. The idea itself (using light to pattern crystallisation) is not novel, although the authors have developed a new strategy that delivers much better results than previously reported. The control achieved over CaCO_3 is excellent, but the other systems are much less impressive. I have just a few other comments:

Comment: *I still remain rather unconvinced of the generality/ overall importance of the approach.*

Response: We thank the reviewer for reviewing our manuscript for a second time. To emphasize the generality and versatility of our method, we performed a new set of experiments. Specifically, and also motivated by a comment of reviewer #4, we induced the nucleation and growth of the biorelevant minerals strontium sulfate and calcium phosphate using NIR laser heating (Figure R5). Similar to the results with carbonate salts, we can induce nucleation by locally heating the solution using NIR and control the growth of single crystals. Note that the Raman spectra of the calcium phosphate salt are in good agreement with the bone mineral hydroxyapatite. Overall, these results demonstrate that our strategy to control nucleation and growth is not limited to carbonate salts and can readily be extended to other minerals that have extensively been studied for their biological importance. We added these new results in the main text, abstract and title, Figure 4, and the Method section and SI with further details.

Figure R5. NIR induced laser induced nucleation and growth of (A) strontium sulfate and (B) calcium phosphate. (i-iii) Optical microscopy timelapse series showing the nucleation and growth. (iv) SEM micrograph, and (v-vi) EDS maps of relevant elements of the resulting minerals.

Comment: *What is the relative importance of the heating of the water and the heating of the ITO? How did the data compare for the calcite experiments if glass was used rather than ITO? Why was glass rather than ITO used in the experiments generating vaterite and aragonite?*

Response: Motivated by the comments of the reviewers, we quantified the heating of the solution as function of the NIR laser power using both glass and ITO as substrate. We found that the solution with the ITO substrate heats up more than the glass substrate, which is consistent with the high IR absorption of ITO. During crystallization, ITO is favorable over glass in two ways: (1) the higher local temperatures on ITO increases the local supersaturation; (2) the higher surface free energy of ITO makes spontaneous nucleation less favorable. We used glass for the vaterite and aragonite experiments for historical to avoid too much heating and thereby maintain a low rate of nucleation. We added the new results on temperature measurements and crystallization behavior on ITO and glass in the manuscript.

Comment: *In my previous review I queried why some reaction solutions were equilibrated for hours prior to initiating crystallisation. Why was this? What was the significance of the length of the equilibration time?*

Response: We observed in the first experiments that (freshly prepared) solutions purely made of CaCl₂, Na₂CO₃, NaOH adjusted, are highly sensitive to undesired spontaneous nucleation. Empirically, we found that these solutions became more stable when left to equilibrate. In a further effort to stabilize the solution we added NaCl, which turned out to stabilize the solution even more, such that no equilibration time was required anymore.

Comment: *I can't find the length of the scale bar in Fig 2C.*

Response: We thank the reviewer for spotting this omission, and incorporated the length of the scalebar.

Comment: *P13 "for light-controlled bottom-up manufacture of highly functional materials..."*

Response: We thank the reviewer for pointing out this error and corrected the text.

Reviewer #4:

The paper by Bistervels et al reports on the use of NIR-based heating to control the nucleation and growth of MCO₃ crystals with spatio-temporal control. This is an important and timely goal and the method proposed is novel. However, this novelty should be better presented and some additional discussion around its rationale and implications would help the manuscript. In general, the manuscript is well-written and presented clearly. In my opinion the manuscript is suitable for publications but should address first the following points.

Comment: *The rationale behind controlling mineralization spatially and temporally could be clearer from the beginning. The first sentence of the introduction suggests this importance but it would help to elaborate and give specific examples....for example the resulting hierarchy dental enamel allows XXX, the organization in nacre allows YYY, etc. Also, in this paragraph, it would be good to also incorporate and mentioned studies using topographies as a technique to guide mineralization.*

Response: We thank the reviewer for these suggestions, and incorporated specific examples to highlight the importance of control over mineralization. Moreover, we also included topography as a method for guiding mineralization.

Comment: *Figure 1B should provide more detail. The experimental setup is an important contribution of the study so the setup should be presented in more detail.*

Response: We now incorporated a detailed figure of the setup in Figure 1B.

Comment: Another point to improve relates to the underlying mechanism of single crystal nucleation and growth for the different polymorphs, particularly explaining the role of fluctuating laser power in controlling nucleation and crystal growth processes. Here, a figure describing this underlying mechanism of mineralization/patterning at the molecular/nano scale would strengthen the manuscript.

Response: Following the reviewer's advice we modified Figure 3, which now includes a schematic on the time-dependent laser powers to control the nucleation and crystal growth processes and description in the figure caption.

Comment: Page 5, the use of additives is mentioned but it is not clear what additives, their concentrations, and their role in MCO₃ crystallization. Please elaborate.

Response: To support the growth of aragonite we complemented the growth solution with 2 mM MgCl₂. For the growth of vaterite and calcite we used the 300 mM NaCl. These details are now clarified in the main text.

Comment: Page 6, authors mentioned that the use of a high-power laser (180 mW) to induce nucleation and later reduced laser power (120 mW) to precisely grow single crystal and to avoid new nucleation. It is not clear why high-power laser won't create multiple nuclei in the first instance during nucleation. Please discuss. Is there a threshold of laser power for preventing new nucleation?

Response: To form a single crystal we keep the nucleation rate low by using very low precursor concentrations, and then use high laser power (180 mW) to induce nucleation. The inherent stochastic nature of nucleation makes it difficult to identify a clear threshold for laser power to prevent new nucleation. However, we do find that the modulation of the laser power is sufficient to separate nucleation and growth for all the systems that we investigated. To clarify these points, we now added a discussion on the combination of concentrations and laser power to the main text.

Comment: Localized heating using electricity has been demonstrated to generate single crystals perovskite in the past (10.1021/acs.cgd.2c00833). The authors should discuss similar studies and directly differentiate themselves. It is not clear how the authors monitored this localized temperature change upon NIR exposure? What was the set up used for monitoring? This should at least be mentioned in the methods section.

Response: We thank the reviewer for this suggestion. Following comments of reviewer #1 and #2, developed a method for locally determining the temperature (see comment reviewer #1 Fig. R1). We added the reference and differentiated our method from previous work in the conclusion section of the main text.

Comment: Also, localized heating due to vibrational absorption of water in the NIR region has been previously exploited in different applications (10.1021/nn500702g). The discussion should also mention these kinds of studies and highlight the novelty of the current work (control of mineralization, etc).

Response: Following this suggestion, we now highlighted these works in the introduction and further clarify the novelty of the current work in the discussion.

Comment: By controlling the patterning of nucleation and growth, the work also contributes and pushes boundaries in the area of hierarchical mineralization, which continues to be a major challenge in materials science. This is part of the novelty of the study and would be worth discussing by comparing to state-of-the-art (10.1038/s41467-018-07658-0,

10.1126/science.aah6350, etc) particularly around the micro-to-macro scale control and connecting back to idea of organisms in nature...eg 10.1002/adhm.201800178, etc.

Response: We agree with the reviewer and now added a more elaborated discussion on the potential of NIR induced crystallization for controlling hierarchical mineralization and included the suggested references in the conclusion section.

Comment: The patterning reported is novel and an important advancement. However, the study and manuscript would be significantly strengthened if the versatility of the technique is demonstrated for example using at least one of the systems mentioned on page 13 (calcium phosphate, nanoparticles assembly, and/or perovskites).

Response: Following the suggestion of this reviewer, and reviewer #3, we now induced the nucleation and growth of the biorelevant minerals strontium sulfate and calcium phosphate using NIR laser heating (Figure R6). Similar to the results with carbonate salts, we can induce nucleation by locally heating the solution using NIR and control the growth of single crystals. Note that the Raman spectra of the calcium phosphate salt are in good agreement with the bone mineral hydroxyapatite. These results demonstrate that our strategy to control nucleation and growth is not limited to carbonate salts and can readily be extended to other minerals that have extensively been studied for their biological importance. We added these new results in the main text and Figure 4, and updated the Method section and SI with further details.

Figure R6. NIR induced laser induced nucleation and growth of (A) strontium sulfate and (B) calcium phosphate. (i-iii) Optical microscopy timelapse series showing the nucleation and growth. (iv) SEM micrograph, and (v-vi) EDS maps of relevant elements of the resulting minerals.

Comment: In the discussion or conclusion there should be some mentioning of potential applications of the patterned MCO3 crystals.

Response: Following this advice, we have now incorporated an outlook on application potential NIR induced crystallization in the conclusion section.

Comment: Please revise minor consistency errors such as scale bar format (Fig 2C), line on top of page 8, etc.

Response: We thank the reviewer for noticing these errors and corrected them.

Reviewers' Comments:

Reviewer #1:

Remarks to the Author:

I want to thank the authors for conducting a new set of experiments where the temperature of the solution at/around laser spot are measured. I think this is rather relevant for groups looking to reproduce these results. I am mostly satisfied with the explanations/clarifications provided by authors. I recommend publication of this manuscript after minor issues listed below are modified.

1) The authors avoid any discussion related to how the measured temperature will be converted to supersaturation(S), which is the key thermodynamic parameter in nucleation & growth. I think this is still missing. There is not a single quantitative S value in the manuscript. Is there a reason why this is the case? At least a discussion on how this can be calculated from Debye-Huckel theory and solution chemistry is needed.

2) The spatial-resolution of the temperature measurements are dictated by the pixel size. I see a scale bar but the size of pixel is not given or I overlooked it. Can you please provide it?

3) There is a contradiction in the explanation of the bubble formation. The authors suggest that the bubbles form due to boiling where the temperature has to be above 100C but the the temperature spot size is below 100 in NIR laser powers used in Figure 1D. This needs clarification. How does their solution boil below 100C at 1 bar with added inorganics.

4) After 17 hours of NIR heating one would expect the temperature of the solution to increase. Does it? Figure 1D only looks upto 300s. Can the authors provide a scaling argument or measurement to address this?

5) As the laser spot is hot and the bulk is cold, you would expect thermally driven flow. Does this play a role in growth of the crystal?

Reviewer #2:

Remarks to the Author:

Adding the measurements of temperature fields in the solution made the manuscript much more convincing. These measurements show that the potential resolution for controlled nucleation is in the order of 100 microns (corresponding to the size of the spot of increased temperature. I find this a very nice paper and have no further comments.

Reviewer #3:

Remarks to the Author:

The authors have made a significant effort to address the comments from all reviewers, and have added significant amounts of new data to their manuscript in the process.

Based on this I am happy to recommend publication in Nature Communications.

Reviewer #4:

Remarks to the Author:

The authors have improved the manuscript, including useful new data, and adequately addressed most of my main comments. However, some have not been addressed properly. Also, please, in your next response to reviewers document, please state the precise location of where the changes are made and paste specific text that addresses the comment.

The writing can be improved. For example, the first sentence of the introduction should be split in two (the content is good as it provides tangible context to the study but is too long and difficult to follow). It would also help to more specifically call references 1-11 to be more precise on how each one supports the different features. Also, I would recommend polishing the text to use more objective and clear language to avoid words such as "tremendous".

Figure 1A-C could be made clearer. The authors have expanded on the information around the equipment but not on the setup and actual process. This can be considerably improved and more clearly presented pictorially, which would be helpful for non-experts to understand/visualize the process. As example, please see Figures 2, 3, and 4 of Yang et al, <https://doi.org/10.1002/admt.202101309>. The information (illustrations) in this figure can then be used in Figures 2b, 2d, 3b, etc to visually connect the process throughout the manuscript.

NCOMMS-23-09486A “Light-controlling nucleation, growth and patterning of biorelevant crystals using resonant near-infrared laser heating”

We would like to thank the reviewers for their time and detailed and constructive comments, which we address point-by-point below. The reviewers’ comments are pasted below in italics while our response is in regular font. Changes in the manuscript are highlighted in yellow.

Reviewer #1:

Comment (1) *The authors avoid any discussion related to how the measured temperature will be converted to supersaturation(S), which is the key thermodynamic parameter in nucleation & growth. I think this is still missing. There is not a single quantitative S value in the manuscript. Is there a reason why this is the case? At least a discussion on how this can be calculated from Debye-Huckel theory and solution chemistry is needed.*

Response (1) We agree with the reviewer that for detailed studies on crystallization mechanism of specific crystal structures of specific chemical compositions it could be relevant to quantify the supersaturation, but such specific cases are beyond the scope of the current work. To still highlight this point, we added the following sentence to the outlook section:

“We foresee that NIR induced crystallization may also be uniquely suitable for quantitatively controlling the local supersaturation, and studying the role of local heating on crystallization dynamics and chemical reactions with spatiotemporal control.”

Comment (2) *The spatial-resolution of the temperature measurements are dictated by the pixel size. I see a scale bar but the size of pixel is not given or I overlooked it. Can you please provide it?*

Response (2) We thank the reviewer for this remark. Each pixel is 0.375 μm , we now included this in the capture of Figure 1Diii:

“Temperature map around the NIR focal point (indicated by black dashed circle) with 180 mW light intensity (pixel size 0.375 μm)”

Comment (3) *There is a contradiction in the explanation of the bubble formation. The authors suggest that the bubbles form due to boiling where the temperature has to be above 100C but the the temperature spot size is below 100 in NIR laser powers used in Figure 1D. This needs clarification. How does their solution boil below 100C at 1 bar with added inorganics.*

Response (3) Figure 1D is measured on glass while Figure 1E-G are measure on ITO coated glass. We now clarify this on the caption of these figures. Note that the irradiated part of the ITO coated substrate will be even higher in temperature than the measured solution due to additional heating (see SI, Fig. S3). For clarification, we added this statement in section 1 of the SI.

Comment (4) *After 17 hours of NIR heating one would expect the temperature of the solution to increase. Does it? Figure 1D only looks upto 300s. Can the authors provide a scaling argument or measurement to address this?*

To measure the bulk temperature upon long NIR irradiation, we placed a thermocouple ca. 1mm from the focal spot. Using the thermocouple, the bulk solution measured at ca. 1 mm from the heating spot rises only from ca. 26 °C to 30 °C during 17 hours of NIR heating. We added this statement to the SI in section 1.

Comment (5) *As the laser spot is hot and the bulk is cold, you would expect thermally driven flow. Does this play a role in growth of the crystal?*

We indeed observe flow during heating. This can play a role in the crystallization by favoring mass transfer. To point this out, we added the following statement to the main text:

“Notably, we observe flow towards the focal spot, which can further support local crystallization by mass transport.”

Reviewer #4:

The authors have improved the manuscript, including useful new data, and adequately addressed most of my main comments. However, some have not been addressed properly. Also, please, in your next response to reviewers document, please state the precise location of where the changes are made and paste specific text that addresses the comment.

Comment (1) *The writing can be improved. For example, the first sentence of the introduction should be split in two (the content is good as it provides tangible context to the study but is too long and difficult to follow). It would also help to more specifically call references 1-11 to be more precise on how each one supports the different features. Also, I would recommend polishing the text to use more objective and clear language to avoid words such as “tremendous”.*

Response (1) We thank the reviewer for the suggestions. We split the first sentence of the introduction in two parts and specifically called out the reference. This now reads:

“Controlling crystallization at the right place and at the right time is at the core of how organisms organize minerals into fascinatingly complex architectures with superb performance.¹⁻⁶ For instance, biominerals of calcium carbonate and calcium phosphate crystallize into hierarchical structures such as dental enamel,⁷ and nacre for creating materials with extreme resilience and hardness,⁸ and crystallographically well-ordered calcite crystals in starfish tentacles for compounded vision.⁹⁻¹¹”

We also checked the text for objective and clear language and made the following alternations:

“Gaining such spatiotemporal control over crystallization offers exciting opportunities for probing, steering, and ultimately understanding crystallization phenomena in organisms, and opens tremendous opportunities for the development of next-generation artificial materials with advanced functionalities.¹²⁻¹⁸”

is changed to:

“Gaining such spatiotemporal control over crystallization offers opportunities for probing, steering, and ultimately understanding crystallization phenomena in organisms, and new opens

routes for the development of next-generation artificial materials with advanced functionalities.^{12–18},

and:

“Furthermore, tuning the NIR wavelength enables resonant heating for a wide range of solvents, which greatly extends the application potential of NIR induced crystallization towards a broad class of highly functional materials with retrograde solubility.”

is changed to:

“Furthermore, tuning the NIR wavelength enables resonant heating for a wide range of solvents, which extends the application potential of NIR induced crystallization towards a broad class of desirable functional materials with retrograde solubility.”

Comment (2) Figure 1A-C could be made clearer. The authors have expanded on the information around the equipment but not on the setup and actual process. This can be considerably improved and more clearly presented pictorially, which would be helpful for non-experts to understand/visualize the process. As example, please see Figures 2, 3, and 4 of Yang et al, <https://doi.org/10.1002/admt.202101309>. The information (illustrations) in this figure can then be used in Figures 2b, 2d, 3b, etc to visually connect the process throughout the manuscript.

Response (2) Following the reviewer’s advice we modified figures 1B, 1C, and 2B to improve clarity and visually connect the process.

Figure 1:

Figure 2:

Reviewers' Comments:

Reviewer #1:

Remarks to the Author:

My concerns are mostly addressed.

Reviewer #4:

Remarks to the Author:

The authors have addressed my comments. The writing and figures are clearer and more informative.

I also appreciate the effort optimising the response to reviewers document.

This is an exciting study and the manuscript now does justice to it. I am happy for it to be published now.